# Can Plants Perceive Human Gestures? Using AI to Track Eurythmic Human–Plant Interaction

**DOI:** 10.3390/biomimetics9050290

**Published:** 2024-05-12

**Authors:** Alvaro Francisco Gil, Moritz Weinbeer, Peter A. Gloor

**Affiliations:** 1MIT Center for Collective Intelligence, Cambridge, MA 02142, USA; 2School of Computer Systems Engineering, UPM Technical University of Madrid, 28031 Madrid, Spain; 3Biodynamische Ausbildung Schweiz, 8462 Rheinau, Switzerland

**Keywords:** plant–human interaction, signal processing, plant action potentials, machine learning, eurythmy, plant biosensors

## Abstract

This paper explores if plants are capable of responding to human movement by changes in their electrical signals. Toward that goal, we conducted a series of experiments, where humans over a period of 6 months were performing different types of eurythmic gestures in the proximity of garden plants, namely salad, basil, and tomatoes. To measure plant perception, we used the plant SpikerBox, which is a device that measures changes in the voltage differentials of plants between roots and leaves. Using machine learning, we found that the voltage differentials over time of the plant predict if (a) eurythmy has been performed, and (b) which kind of eurythmy gestures has been performed. We also find that the signals are different based on the species of the plant. In other words, the perception of a salad, tomato, or basil might differ just as perception of different species of animals differ. This opens new ways of studying plant ecosystems while also paving the way to use plants as biosensors for analyzing human movement.

## 1. Introduction

Aristotle, in his hierarchy of all matter and life, puts humans (of course) at the top of the evolutionary pyramid, as possessed by reason, soul, and language. Animals are distinguished from plants in their ability to move and sense their surroundings. Plants, in Aristotle’s categorization, lack movement and the ability to sense. Aristotle further distinguishes between different levels of soul. The vegetative soul is on the lowest level, and it is responsible for basic functions like growth and nutrition. The sensitive soul of animals adds perception and movement. On the highest level is the rational soul, unique to humans, which enables reason, thought, and language. While anybody speaking to houseplants has been questioning this view of the vegetative soul of plants for a long time, plant biologists until very recently have still subscribed to variations of this approach by considering plants basically automatons with no indications of a sensitive soul. One notable exception is Charles Darwin, who in his work *On the Variation of Animals and Plants under Domestication* published in 1868, already mentioned the possibility of plants having some level of memory or sensitivity [1]. Darwin’s suggestion was taken up by Jagadish Bose (1906) by developing innovative instruments like the crescograph to measure minute plant movements and responses to stimuli, which allowed him to gather previously unattainable data. He documented plant responses to light, touch, chemicals, and electrical currents, suggesting some level of communication and interaction. He proposed a “nervous system” for plants [2], formulating the “pulsatory theory” suggesting a quasi-nervous system in plants that was different from animals. This challenged the prevailing view of plants as static beings. Like Darwin, Bose avoided claiming sentience, but he argued for recognizing plants as sensitive and responsive organisms demonstrating intelligence in their own way.

In our research presented in this paper—inspired by Bose—we have been measuring the voltage differentials of plants in response to human movements. In particular, we conducted a series of experiments, where changes in voltage over time in response to eurythmic movements were collected, documenting different electric responses of the plants to different eurythmic gestures. Eurythmy is a movement art that translates sound and language into expressive bodily gestures (Figure 1). It is often used in performance, education (especially Waldorf schools), and in therapy.

The contributions of this paper are threefold. First, we introduce a novel method to measure human movement using plants as biosensors by tracking the voltage differential between roots and leaves of plants. Secondly, by building machine learning models, we demonstrate that plants are capable of distinguishing between different types of human movements. We do this through tracking different types of eurythmic movements. Thirdly, we show that the responsiveness to human movement using our approach differs between different species of plants, investigated with three vegetable species, lettuce, basil, and tomatoes.

## 2. Related Work

Plants, like animals, use electrical signaling as a means of communication and coordination within their tissues and organs. While the mechanisms of electrical signaling in plants are not as well understood as those in animals, research has uncovered important aspects of how it operates [3]. Plants can generate electrical signals called action potentials, which are rapid changes in the electrical potential across the cell membranes of specialized cells called “plant neurons” or “electrogenic cells” [4]. These action potentials are similar to those found in animal neurons but are generally slower and less pronounced. Action potentials in plants propagate from one cell to another through plasmodesmata, which are cytoplasmic channels that connect adjacent plant cells. This allows electrical signals to travel over long distances within the plant. Various stimuli trigger electrical signaling in plants, including mechanical stimuli (e.g., touch, wounding, or wind), environmental stimuli (e.g., changes in light, temperature, or humidity), and chemical stimuli (e.g., hormones or signaling molecules). Electrical signals in plants elicit a wide range of physiological responses, such as leaf movements, changes in growth patterns, and the synthesis of defensive compounds [5]. These responses help plants adapt to their environment and protect themselves from potential threats.

There are several active areas of research aimed at better understanding electrical signaling and communication mechanisms in plants. Researchers are using electrophysiological techniques, such as patch-clamp methods and voltage-sensitive dye imaging, to study the biophysical properties of action potentials in plant cells and the ion channels involved in generating and propagating these signals. Another line of research studies the genes and molecular components involved in plant electrical signaling pathways. This includes exploring the roles of various ion channels, receptors, and signaling molecules in the generation and transmission of electrical signals [4]. There is growing interest in exploring the similarities and differences between electrical signaling in plants and animals, giving rise to the field of plant neurobiology. This research aims to understand the evolutionary origins and functional implications of electrical signaling in plants [6].

Recent research focuses on understanding how plants sense and respond to their environment, proposing a plant neurobiology network independent of a nervous system [7]. Mancuso and Viola explore how plants communicate with each other and their surroundings through electrical signals, chemicals, and even sound waves [8].

Lately, researchers have challenged traditional views of plants as passive organisms, shifting toward recognizing the complexity and potential intelligence hidden within the plant kingdom. They propose that plants exhibit some sort of self-awareness different from animals. They found that plants emit and respond to various signals, including sound, chemicals, and electrical signals [9,10,11]. This suggests potential communication and information exchange between plants. Studies have shown that plants can adjust their behavior based on past experiences, suggesting a form of learning and memory [9]. It is also argued that plants demonstrate decision-making abilities, for example, in resource allocation based on environmental cues [12].

In plant signal analysis, the utility of feature extraction and machine learning is supported by several pioneering studies that demonstrate the technique’s adaptability to the vegetative domain and highlight the sensitivity of plants to a range of stimuli. In the following studies, feature modeling was used to recognize the electrical responses of the plants, to their internal state like water stress and circadian rhythms [13], but also to external stimuli like different frequencies of sounds [14]. Further research has leveraged this technique to illuminate plant–human interaction, differentiating the plant reaction to different humans and moods [15], and discovering the presence of eurythmic human gestures [16]. These collective findings underscore the usefulness of feature extraction in revealing the complex, responsive nature of plants to both abiotic and biotic stimuli, opening new avenues for understanding vegetative perception and communication.

## 3. Hypothesis

This paper investigates three hypotheses analyzing the electrical reactivity of plants and their interaction with human movements through machine learning.

**H1.** 
*Plants serve as effective sensors for detecting human movement.*


The first hypothesis lays the foundation for this work by demonstrating the general responsiveness of plant electrical reactivity to human movement. This hypothesis has been supported by earlier preliminary research [17], which validated the concept with a smaller data sample. We seek to confirm the previous study and measure the impact with increased accuracy using a larger dataset.

**H2.** 
*Plants show different types of responses to different types of human movements.*


This hypothesis investigates plant sensitivity to variation in human activity beyond mere presence or absence of human movement. This will be investigated using machine learning to predict distinct eurythmy gestures by electric potential difference patterns plants.

**H3.** 
*The electrical signals of different plant species (salad, basil, tomatoes) contain unique characteristics that allow for the differentiation between plant species.*


This hypothesis investigates if there are inherent properties in the plant’s electrical response that differ between different plant species.

## 4. Method

### 4.1. Experimental Setup

The ultimate goal of our research is to develop an electrotechnical measurement method that allows us to track the influence of humans on plants in their interaction with the plant. In this project, we investigate the reaction of three plant species (iceberg lettuce, basil and tomato) to eurythmy gestures by measuring the electrical potential in order to gain a deeper understanding of how and at what point human movement interacts with the plant. The three plant species—iceberg lettuce, basil and tomato—were each planted in three differently treated plots on an approx. 3.5 m wide and 75 m long planting strip in the Swiss Biodynamic Education Research Garden in Rheinau. The distance between the individual plots of the same plant species was at least 10 m. An artificial meadow was between the individual plots, or there were plots of the other plant species. The meadow was mowed regularly to avoid creating shadow effects. Over the vegetation period, we sequentially planted four generations of lettuce and two generations each of basil and tomatoes. In order to exclude the possible influence of abiotic factors (position of the sun, soil conditions, weather influences, etc.), we randomized the distribution and orientation of the three plots across the field strips in the generations. The three plots were treated as follows:In one section of the field, plants received eurythmy treatments during each day of measurement;In a different section, plants were organized in rows that were treated with eurythmy for the first and only time;In a third location, the control group was planted, which was never treated with eurythmy.

Using the plant SpikerBox [18], we measured the electrical potential of the plant between the soil and the leaf with the help of electrodes and observed the change of potential during the treatment of the plant with eurythmy (Figure 2). The Spike Recorder software stored the recorded electrical potentials in wav files.

Two of the above-mentioned groups of plants (“regular eurythmy” or “one-time eurythmy” together with the “untreated control group”) were measured simultaneously with the eurythmy being performed by the same person (Figure 3). The eurythmy gesture series were varied by plant type to randomize the effects: “A-G-D” gestures for lettuce, “A-G-D-O” gestures for tomatoes, and “A-G-D-L” gestures for basil, each repeated four times per plant. For each performance, six plants were recorded at the same time: three in the control group and three undergoing eurythmy. To align the different eurythmy gestures with the electric plant readings, we also recorded the person conducting the eurythmy using a camera. The measurements were carried out weekly where possible, depending on the weather conditions. We took the “A-G-D” measurements universally for all three plant species, and these are the ones analyzed in this paper. The additional measurements for “O” and “L” are reported here just for the sake of completeness; they will be used in further work.

During one vegetation period, it is possible to grow four generations of lettuce but only a maximum of two in basil and tomatoes, respectively. However, as we did not compare “lettuce” results to those of “basil” and “tomatoes”, but treated them as between-species-independent measurements, it is only a matter of how many independent data points we could sample within each species. We took 46 measurements on lettuce, 30 on basil and 31 on tomato, with half of the measurements forming pairs of “regular eurythmy” with “control” and the other half forming pairs from “one-time eurythmy” with “control” (Figure 4).

This structured approach to sampling across different plant species and generations under eurythmy treatment schedules provided a comprehensive dataset for analyzing the impact of human movement on plant electrical responses. In total, the following measurement data were collected (Table 1):

### 4.2. Data Preparation

#### 4.2.1. Synchronization

In the initial phase of data preparation, we addressed the issue of non-simultaneous recording by trimming a few seconds from the beginning and end of each file. This ensured all recordings matched in length and corresponded to the same experiment time period.

#### 4.2.2. Standardization

In alignment with precedents set by previous research, our normalization technique involves adjusting for environmental and sensor placement variances by standardizing signal data [19]. Specifically, for each plant data file, we normalize by subtracting the mean and dividing by the standard deviation. This approach not only addresses the relative nature of the SpikerBox’s signal output, as specified by BackyardBrains [20], but also effectively reduces the influence of environmental variables on signal amplitude.

#### 4.2.3. Data Point Selection

Based on the video recordings, human labelers marked the precise moments each eurythmy gesture commenced and finished in the wav files (Figure 5). Consequently, every segment of the waveform corresponding to a specific eurythmy gesture was taken as an individual data point. Given the experimental setup, each plant species received a distinct sequence of eurythmy gestures: lettuce was exposed to “A-G-D” four times per session (12 gestures total), while tomatoes and basil were each given “A-G-D-O” and “A-G-D-L”, respectively, also four times, amounting to 16 gestures per session. From the 270 lettuce, 182 tomato, and 173 basil recordings, we constructed a dataset of (12 × 270) + (16 × 182) + (16 × 173) = 3240 + 2912 + 2768 = 8920 data points. However, due to a few gestures not being correctly recorded, the cleaned dataset is composed of 8878 data points.

To explore whether plants exhibit measurable responses to nearby human movements, we applied the data point selection methodology of a previous study [17]. During eurythmy performances, six plants were recorded simultaneously, out of which three were exposed to human movement. This approach ensured our dataset of 8878 data points was well balanced, comprising 4483 eurythmy and 4395 control samples (Table 2), enabling a robust analysis of plant reactions to human presence and absence (control).

To explore if plants respond differently to different eurythmy movements, we had to narrow our focus to the subset of plants directly exposed to eurythmy, totaling 4483 data points. To ensure a comprehensive analysis, we concentrated on the gestures present across all three plant species studied: A, G, and D. Consequently, this focus resulted in a reduction of 388 letters “O” and 324 letters “L”, leaving us with 3771 data points for conducting a detailed 3-class classification (Table 3).

Finally, we also explored whether specific plant species exhibit distinctive electrical potential characteristics. For this inquiry, we used the complete dataset, as we assumed the influence of human presence to be a separate dimension not relevant for this analysis. Instead of trimming data based on eurythmy gestures, we opted for a uniform trimming interval of 1 s to increase the number of samples. With 625 wav files averaging 238 s each, our analysis encompassed a final dataset of 148,682 data points, equating to one data point per second of recording (Table 4).

#### 4.2.4. Automatic Detection of Measurement Errors—Introducing the Flatness Ratio

In analyzing the signal dataset, we noticed waveforms occasionally exhibiting flatness, signaling periods where electrical potential remained constant. With wav recordings capturing 10,000 samples per second, such flat segments might indicate that the plant’s electrical potential does not vary at the same rate as the sampling, or—more likely—may reflect issues with the SpikerBox sensor’s functionality. We therefore developed a metric to identify outliers by computing the degree of flatness (Appendix A).

The key aspect of the problem consists of determining the appropriate threshold of consecutive samples of identical values that indicate a measurement error rather than natural variability in the plant’s electrical potential. While a span of 1–100 samples (0.0001–0.01 s) remaining constant is within expected fluctuations, sequences of 1000–10,000 constant samples (0.1–1 s) suggest probable sensor malfunctions.

The flatness ratio metric, designed to analyze signal datasets, quantifies the extent of waveform flatness by identifying prolonged periods where electrical potential remains unchanged. This function calculates the ratio of segments within a signal where consecutive values stay constant beyond a predefined threshold. To comprehensively assess the efficacy of our metric, we experimented with thresholds set at [100, 500, 1000, 5000, 10,000] samples in our datasets.

Upon examining the outcomes of the five distinct metrics derived from varying thresholds, we established a hierarchical rule that effectively identifies errors within our dataset through visualization and analysis of their distribution. This rule applies if the flatness ratio at a 1000 sequential samples threshold (flatness ratio 1000) exceeds 75%, coupled with a flatness ratio at a 500 sequential samples threshold exceeding 75%, and flatness ratio 100 surpassing 99.9% (Table 5). For a practical demonstration of this rule’s application, Figure 6 showcases its effectiveness in identifying the second wave as an outlier.

### 4.3. Feature Extraction

Feature extraction is a critical step in processing complex data for machine learning. This technique is also prevalent in sound wave analysis, where the extraction of features such as Mel-frequency cepstral coefficients (MFCCs) [21] or timbral characteristics has significant prediction accuracy. For instance, the work by Tzanetakis and Cook on music genre classification [22] underscores the adaptability of feature extraction techniques across varied and complex datasets. Informed by these precedents, we adopted feature extraction to transform variable-length waveforms, as encountered with the different eurythmy gestures, into uniform-length feature vectors.

For most features, we had to convert the waveforms from their original temporal representations into the frequency domain. A straightforward conversion of the full wave into frequency representation is impractical, as it predominantly generates noise, obscuring the signal’s informative aspects. The established solution, widely documented in both academic literature and applied methodologies, involves windowing. In the context of our investigation, we combined the resulting feature vectors from each window into a singular, aggregated feature vector that accurately reflects the signal in its entirety. Aggregation was achieved by calculating the mean and standard deviation of the features for every window.

In the process of windowing, two crucial parameters need selection: window size (ws) and hop length (hl). Window size refers to the duration of audio signal segments analyzed for frequency content at one time, impacting the detail of the frequency analysis. A larger window captures more of the audio signal for a richer frequency analysis but can blur quick changes. Hop length is the distance between the start points of consecutive windows, controlling the amount of overlap between them. It influences how frequently the signal is sampled for analysis, affecting the resolution of the time series analysis. We experimented with various values for these parameters to find the optimal balance for our analysis (Table 6).

The Mel-frequency cepstral coefficients (MFCCs) computation typically involves transforming the frequency to the Mel scale, which better mirrors human hearing. In our work, we opted to compute cepstral coefficients with Librosa [23], intentionally excluding the Mel-scale transformation. This choice reflects our understanding that for electrical signal analysis, the conventional Mel scaling may not optimally align with our goals, as it is optimized for human hearing rather than for plants.

We employed a variety of libraries to compute features (Table 7), including Librosa [23], PyAudioAnalysis [24], Numpy [25], and PyEEG [26], which were each selected for their specialized capabilities in processing distinct aspects of our data. Librosa cepstra mean and deviation (features 1–26) were calculated from the waveform by first computing the Short-Time Fourier Transform (STFT) and converting it to a log scale to obtain the cepstrum. From this, the cepstral coefficients were selected, and the mean and standard deviation were then calculated across all windows, producing a set of cepstral features that encapsulate both the average spectral shape (mean) and its variability (deviation) over time. In pyAudioAnalysis, the features 27–94 do not represent single values extracted directly from the entire waveform, but also, the mean values aggregated from each window across the signal, providing an analysis of the signal’s characteristics over time. Subsequently, features 95–162 were determined as the standard deviation of these short-term features, offering insights into the variability within each signal’s feature set. Finally, features 163–172 are derived from the entire signal without employing windowing. This approach allows these features to capture overarching characteristics and trends within the signal, reflecting its global properties rather than the moment-to-moment variations highlighted by windowed analysis.

### 4.4. Feature Analysis

In our feature analysis, we started by examining feature disparities between feature groups using the Kruskal–Wallis test to ascertain significance. Given the common occurrence of correlated features in signal feature extraction, which can negatively impact machine learning model efficacy, we performed a feature reduction. This process involved retaining only those features from Table 7 with correlations below a specified threshold, ensuring our model is trained on non-redundant, informative data. To optimize the balance between feature retention and model accuracy, we experimented with various correlation thresholds, including 0.7, 0.8, and 0.9.

### 4.5. Machine Learning

Machine learning (ML) is a subset of artificial intelligence (AI) that equips computers with the ability to learn and improve from experience without being explicitly programmed for specific tasks. This capability is particularly valuable in environments where the volume and variety of data exceed the human capacity for analysis. In the study, we utilized machine learning with the goal of training our model to identify the specific feature combinations indicative of each group. This learning allows the model to discern between the features of waveforms it has never encountered before (test data). Following conventional machine learning protocols, our dataset was randomly partitioned into an 80% training set and a 20% testing set.

To ensure the reliability of our classifications and to deepen our analytical insights, our research employed a varied ensemble of machine learning models from leading libraries such as Scikit-learn [27], LightGBM [28], and XGBoost [29]. Scikit-learn provided the foundation for the creation and training of all our models with the exception of our LGBM and XGB models. For these, we employed LightGBM and XGBoost, respectively. This methodological diversity allowed us to validate our findings across different algorithms and frameworks, enriching our understanding of the dataset’s complexity. Table 8 shows the different machine learning approaches and the hyperparameters we used.

## 5. Results

### 5.1. H1: Do Plants Recognize Eurythmic Gestures?

In our analysis, we identified a range of features that differed significantly between the experimental (eurythmy) and the control group. However, due to correlations among many of these features, we reduced the number of features for the subsequent analysis. Table 9 presents the Kruskal–Wallis test of only those features that are statistically significant and exhibit a correlation coefficient of less than 0.8 with each other.

As shown in Table 9, many features of humans performing eurythmy are significantly different from the control. Features such as the mean, the slope sign changes ratio, and variance are straightforward for interpretation and visualization. To analyze the difference between experimental group (vegetable readings exposed to eurythmy) and control group (vegetable readings of non-exposed plants), we created a canonical representation of the two groups, averaging them over the duration of an eurythmy gesture (Figure 7).

The canonical representation effectively illustrates what the Kruskal–Wallis test indicates: both the mean and the standard deviation within the eurythmy group are lower. The reduced number of oscillations of the curve, i.e., the slope sign changes ratio—how often the wave switches between positive and negative directions—is also lower in the eurythmy group.

In our machine learning analysis, we evaluated the trained models using a “holdout” dataset. As detailed in Table 10, the LightGBM (lgbm) model demonstrated the best performance, achieving an accuracy of 0.749 and an F1 score of 0.748, outperforming the baseline (majority class 0.503) by 49%. These results are further illustrated in the confusion matrix in Figure 8.

### 5.2. H2: Differentiating between A-G-D Eurythmy Gestures

For our second hypothesis, investigating variations in plant reactions between different eurythmy gestures, we identified statistically significant features as well. For the feature reduction, we used a correlation threshold of 0.9; the significant features are detailed in Table 11.

Figure 9 shows again the canonical representation of the potential curve averaged between all samples for each of the three different eurythmic gestures (A, G, D).

Using machine learning to distinguish the three classes A, G, and D, the model outcompeted the baseline (majority class 0.346) by 32%. Gradient boosting performed best with an accuracy of 0.458 and an F1 score of 0.453 (Table 12). The confusion matrix is shown in Figure 10.

### 5.3. H3: Distinguishing Signals of Different Plant Species

For our final hypothesis, investigating variations in plant reactivity between different plant species, statistically significant features were also found. We employed a correlation threshold of 0.8 for the feature reduction; the discerned features are detailed in Table 13.

In our last analysis, comparing the canonical voltage curves of salad, tomato, and basil, we again found noticeable differences in amplitude mean between the canonical waveforms of each species (Figure 11). Due to the same length of all waves in this analysis (1 s), no resampling was required when aggregating the canonical curves.

Achieving an accuracy of 0.508 and an F1 score of 0.505, the XGBoost (xgb) model showcased superior performance in our machine learning experiments (Table 14). This performance denotes a 41% enhancement over the baseline’s majority class accuracy of 0.36, illustrating the model’s capacity to accurately identify distinctive features among the plant species. For a visual representation of the model’s performance, refer to the confusion matrix in Figure 12.

## 6. Discussion

As has been demonstrated by the results in the previous section, utilizing plants as biosensors for detecting human movement through their electrical signals definitively merits further investigation. The statistical tests conducted to explore the different hypotheses underscore the importance of researching a wide range of features. MFCCs and cepstrals differ only by the Mel transformation step, but both of them play distinct roles in interpreting plant signals. MFCCs were found to be the significant for the first hypothesis, distinguishing between eurythmy and control. In contrast, cepstrals were central in the second hypothesis, differentiating between the three gestures for A, G, and D. At the same time, both of them proved crucial for the third hypothesis, distinguishing between salad, tomato, and basil. This result highlights the critical role of feature selection in signal processing and the potential for discovering unique, meaningful features within plant signal datasets. The effectiveness of these different features across hypotheses underscores a key insight: even closely related feature sets can serve divergent analytical purposes, suggesting the presence of underlying complexities in plant response mechanisms that are yet to be fully understood.

Further precision in the discussion is provided by the analysis of the machine learning models’ performance; the best hyperparameters are provided in Appendix B. The model for the first hypothesis achieved an accuracy of 0.749, demonstrating an increase of 49% over the baseline and 16% over the previous analysis (0.63) [17]. These results not only reaffirm the hypothesis that plants can act as effective biosensors for human movement but also support the assertion that adding more data enhances the model’s ability to discern between classes effectively.

The accuracy of 0.458 for the second hypothesis, marking a 32% increase over the baseline, indicates that plant responses to human movement are subtler and more complex than mere presence detection. This complexity suggests that plants can differentiate between different movements, although this task proves more challenging than the first hypothesis indicated. The higher model accuracy in the first hypothesis could be attributed to the presence of more statistically significant features, which simplified the classification task. The reduced accuracy observed for the second hypothesis might result from fewer samples per class, the need to identify new significant features specific to each movement, or the inherent difficulty in distinguishing the plants’ reactions to these movements. A promising direction for future research would be to explore variations in the effusiveness of movement, such as intensity or speed, to determine if these factors elicit stronger reactions from the plants, moving beyond mere movement direction as happens with different eurythmy gestures.

For the final hypothesis, the model’s accuracy rate of 0.503, coupled with a 39% improvement over the baseline, suggests that the reactions of plants to their surroundings exhibit unique characteristics inherent to each species. This finding aligns intuitively with our understanding of individual responses among humans and animals. In exploring this hypothesis, the study shifted focus exclusively to the plants, sidelining the dynamics of plant–human interactions to concentrate on the plants’ intrinsic reactivity to their environment. This approach heralds new methodologies for investigating plant ecosystems, emphasizing the significance of species-specific responses in understanding plant behavior and interactions. Such insights pave the way for deeper explorations into the complexity of plant ecosystems, potentially revolutionizing our approach to studying vegetative life and its nuanced interactions with the surrounding environment.

## 7. Limitations

The results of our study, while promising, are subject to several limitations that warrant careful consideration. Firstly, the issue of causality versus correlation presents a significant challenge. While we observe changes in voltage readings that correlate with human presence or movement, it is not possible from this study alone to definitively determine causation. The observed changes could result from various factors, including direct movement, the electrical field generated by the human body, or even subtle vibrations transmitted through the ground by the person performing the eurythmic gestures.

Secondly, unlike established standards in neuroscience for conducting EEG studies [30], this field lacks a universally accepted protocol for performing equivalent measurements in plants. This gap is compounded by the challenges posed by using the BackyardBrains SpikerBox, which relies on external electrodes susceptible to measurement errors, in contrast to potentially more accurate but invasive internal methods that could harm the plant. We addressed measurement error issues by implementing the flatness ratio, which is a metric designed to systematically identify sensor malfunctions and improve data reliability.

Moreover, the risk of overfitting is a common concern in machine learning endeavors. To mitigate this, we employed a diverse array of models to ensure the robustness of our findings, testing the data’s resilience against overfitting through this multiplicity of analytical approaches.

Our decision to focus on feature extraction was driven by a desire to enhance the explainability of our models. However, it became apparent that the features do not always directly indicate wave-based outcome data, necessitating that models learn non-linear representations of the data to achieve accurate classification. This aspect highlights a tension between the goals of explainability and model performance, underlining the complexity of accurately interpreting plant electrical signals.

Finally, broadening the scope of our project and analyzing more plant species would greatly improve the validity of our approach. As we were quite limited in our resources, and the experiments described in this paper already took a lot of time and effort, we hope that our work will motivate other researchers to replicate and extend our approach with other plant species to demonstrate the broad applicability of this method.

## 8. Conclusions

Our work draws on results from recent research that highlight the synergistic capabilities of machine learning to investigate plant responses to various stimuli, thereby extending our understanding of plant electrophysiology. This study investigated if plants of three different species respond to human movement, employing feature extraction and machine learning to analyze voltage differentials in response to eurythmic gestures. While the results we obtained in this study are encouraging, there are many opportunities for future research. Future studies could, apart from gathering more data, broaden this spectrum by incorporating a wider range of plant species and more detailed human–plant interactions, thus offering more insights into the dynamism of plant electrical responses. Additionally, deep learning approaches will potentially improve accuracy and enable further insights. Despite the challenges in establishing causation and the limitations of current measurement protocols, this research lays the groundwork for future explorations into plant–human interactions and opens new pathways for interdisciplinary research.

## Figures and Tables

**Figure 1 biomimetics-09-00290-f001:**
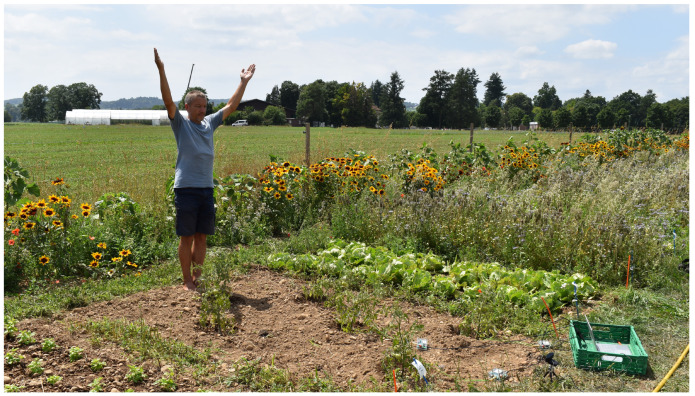
Picture of one experiment performed at the Biodynamic Research Garden, Rheinau, Switzerland.

**Figure 2 biomimetics-09-00290-f002:**
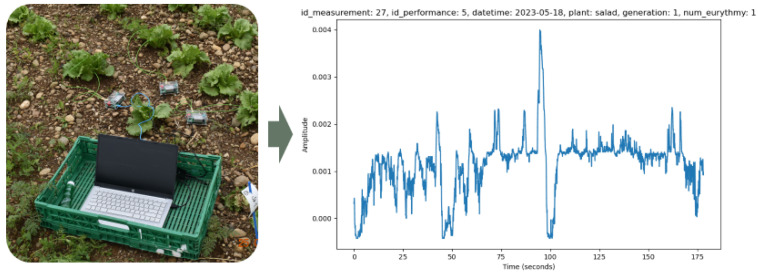
Measurement of voltage differences in lettuce and resulting measurement data.

**Figure 3 biomimetics-09-00290-f003:**
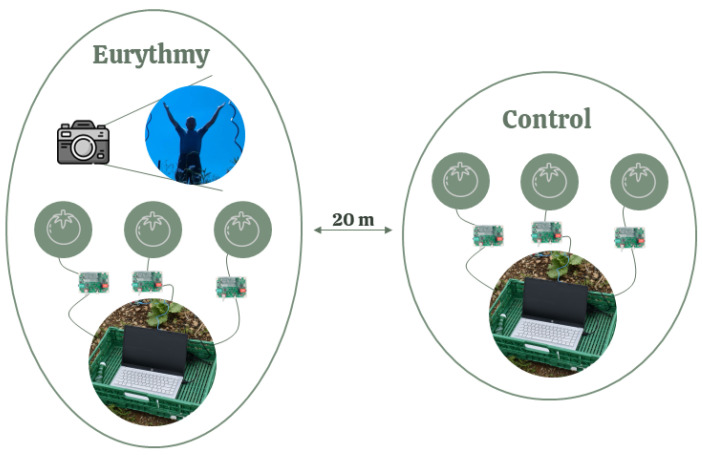
Schematic representation of one measurement.

**Figure 4 biomimetics-09-00290-f004:**
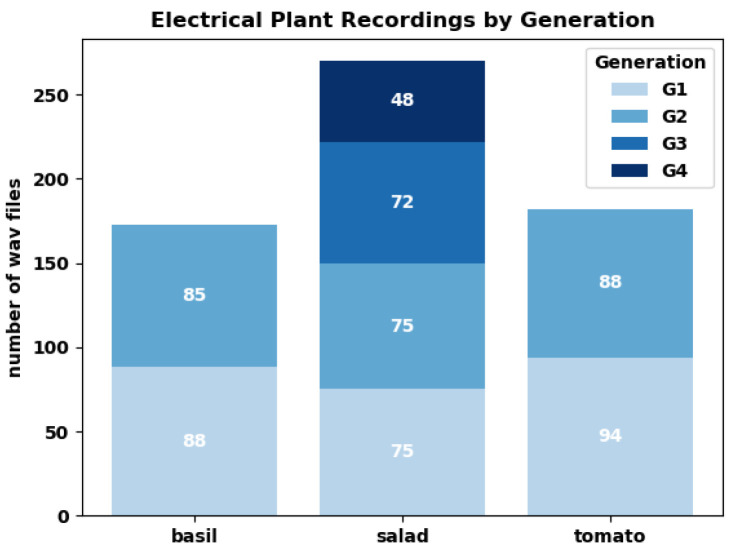
Histogram of plant electrical recordings by generation and plant.

**Figure 5 biomimetics-09-00290-f005:**
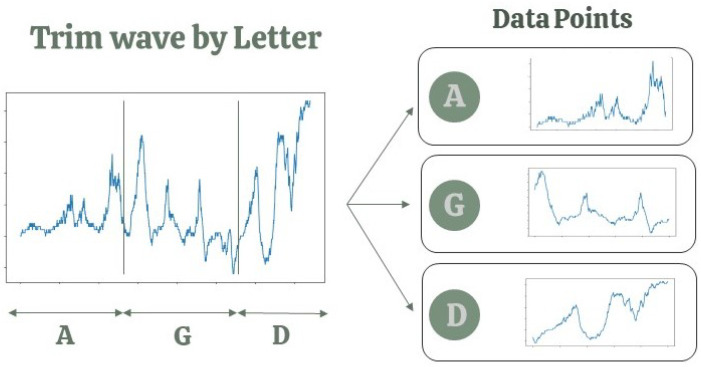
Schematic display of automatic trimming of hand-labeled signals.

**Figure 6 biomimetics-09-00290-f006:**
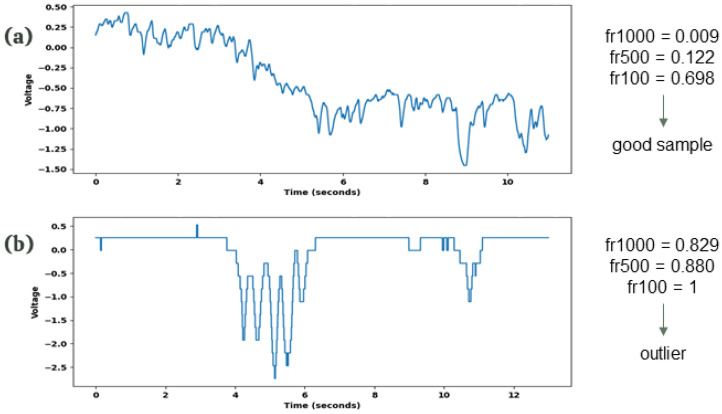
Two different waves from the dataset with their respective calculated flatness ratio metrics. (**a**) Normal wave, with all flatness values below the thresholds. (**b**) Outlier, with all values above the threshold. See the appendix for an explanation of the flatness ratio.

**Figure 7 biomimetics-09-00290-f007:**
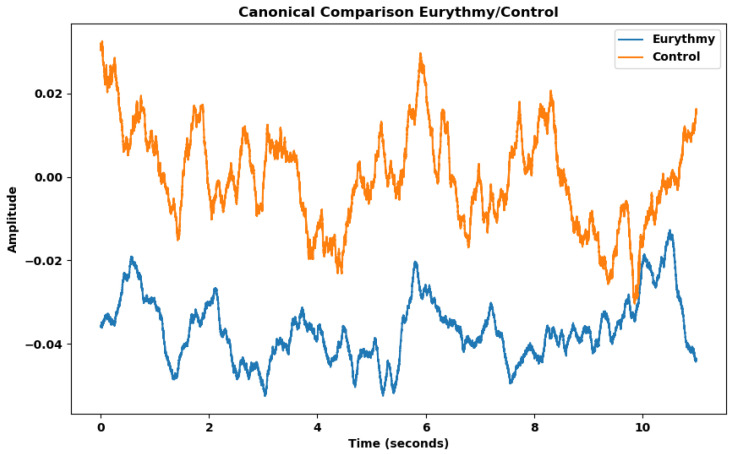
Eurythmy and control average electrical signals when eurythmy gestures were performed. Each line represents the mean of all recorded values for voltage changes over time associated with eurythmy recordings, and control recordings, respectively.

**Figure 8 biomimetics-09-00290-f008:**
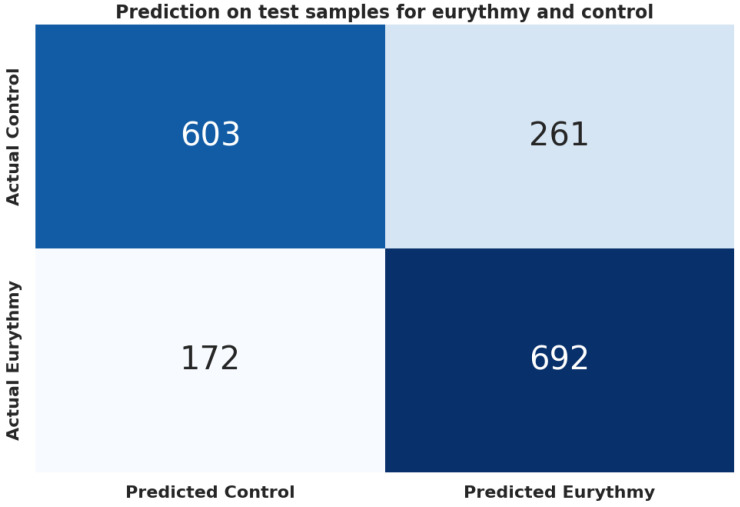
Confusion matrix for the lightGBM (lgbm) model between eurythmy and control.

**Figure 9 biomimetics-09-00290-f009:**
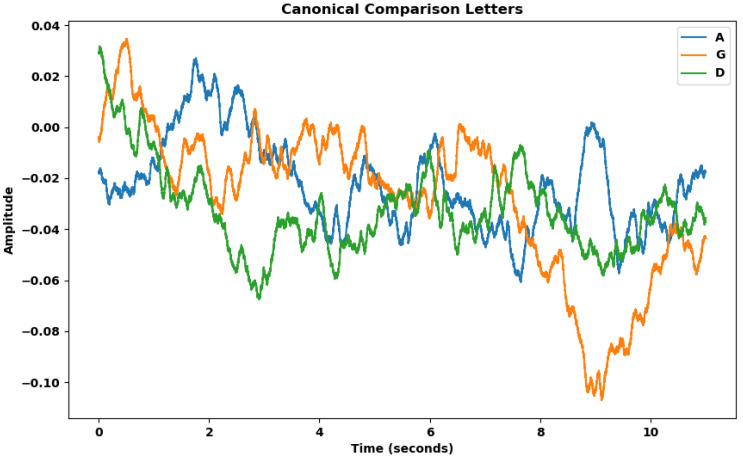
Canonical representation of the average potential curves for each group, derived from resampled and averaged data. Each line represents the mean of all recorded values for voltage changes over time associated with gestures corresponding to the letters "A," "G," and "D," respectively.

**Figure 10 biomimetics-09-00290-f010:**
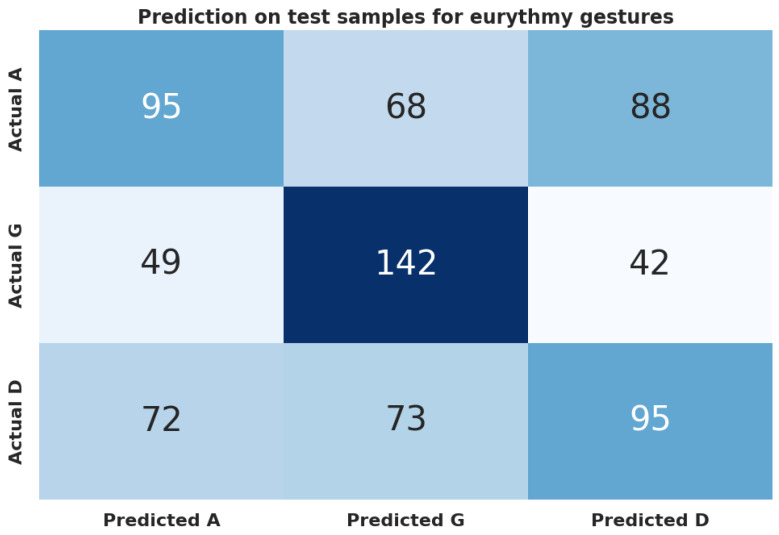
Confusion matrix for the XGBoost (xgb) model between eurythmy letters (A-G-D).

**Figure 11 biomimetics-09-00290-f011:**
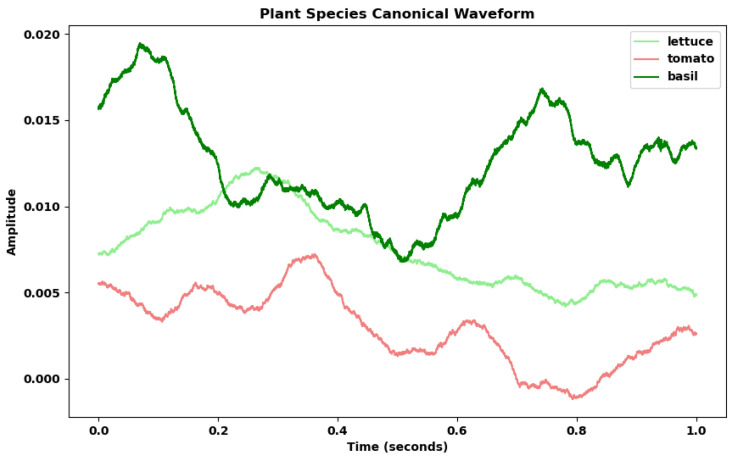
Canonical representation of voltage curves for lettuce, tomato, and basil. Each line represents the mean of all recorded values for voltage changes over time associated with lettuce, tomato, and basil respectively.

**Figure 12 biomimetics-09-00290-f012:**
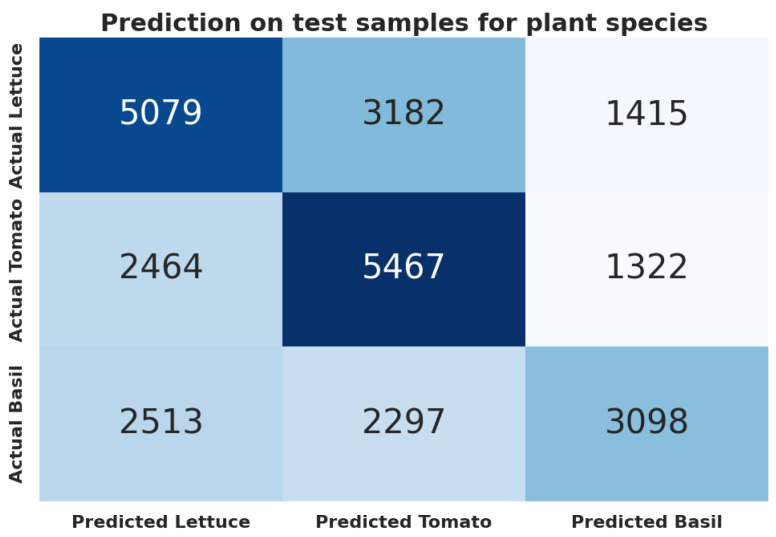
Confusion matrix for the lightGBM (lgbm) model between lettuce, tomato and basil.

**Table 1 biomimetics-09-00290-t001:** Data collection summary.

Plant Type	Electric Recordings (Wav Files)	Eurythmy Performances (Mp4 Files)
Lettuce	270	46
Tomato	182	31
Basil	173	30

**Table 2 biomimetics-09-00290-t002:** Data points for comparing eurythmic response to control.

Group	Data Points
Eurythmy	4483
Control	4395

**Table 3 biomimetics-09-00290-t003:** Data points for comparing the three different gestures A-G-D.

Group	Data Points
Gesture A	1261
Gesture G	1255
Gesture D	1255

**Table 4 biomimetics-09-00290-t004:** Data points for comparing the three plant species.

Plant Type	Data Points
Lettuce	53,966
Tomato	50,141
Basil	44,575

**Table 5 biomimetics-09-00290-t005:** Data points of the different hypotheses after cleaning.

Dataset	Hypothesis	Rule	Data Points	% of the Initial
Eurythmy/control	H1	All initial data points	8920	100%
Cleaned eurythmy/control	H1	fr1000 < 0.75 and fr500 < 0.85 and fr100 < 0.999	8680	97.3%
A-G-D	H2	All initial data points	3771	100%
Cleaned A-G-D	H2	fr1000 < 0.75 and fr500 < 0.85 and fr100 < 0.999	3616	95.8%
One second samples	H3	All initial data points	148,682	100%
Cleaned one second samples	H3	fr1000 < 0.75 and fr500 < 0.85 and fr100 < 0.999	134,182	90.2%

**Table 6 biomimetics-09-00290-t006:** Parameters used in windowing signals.

Hypothesis	Parameter	Values (s)
H1 & H2	Window Size	1, 2
	Relative Hop Length	ws, ws/2
H3	Window Size	0.1, 0.2
	Relative Hop Length	ws, ws/2

**Table 7 biomimetics-09-00290-t007:** List of features extracted from the wave in this study.

ID	Feature	Description	Library
1–13	Cepstra Mean	The average of cepstral coefficients, capturing the overall trend in the spectral envelope of a waveform.	librosa
14–26	Cepstra Deviation	Standard deviation of cepstral coefficients, reflecting the variation in the spectral envelope over time.	
27	Zero Crossing Rate	The rate of sign changes of the signal during the duration of a particular frame.	pyAudioAnalysis
28	Energy	The sum of squares of the signal values, normalized by the respective frame length.	
29	Entropy of Energy	The entropy of subframes’ normalized energies. It can be interpreted as a measure of abrupt changes.	
30	Spectral Centroid	The center of gravity of the spectrum.	
31	Spectral Spread	The second central moment of the spectrum.	
32	Spectral Entropy	Entropy of the normalized spectral energies for a set of subframes.	
33	Spectral Flux	The squared difference between the normalized magnitudes of the spectra of the two successive frames.	
34	Spectral Rolloff	The frequency below which 90% of the magnitude distribution of the spectrum is concentrated.	
35–47	MFCCs	Mel-frequency cepstral coefficients form a cepstral representation where the frequency bands are not linear but distributed according to the Mel scale.	
48–59	Chroma Vector	A 12-element representation of the spectral energy.	
60	Chroma Deviation	The standard deviation of the above 12 chroma coefficients.	
61–94	Delta of Features 27–60	The changes in the values of features 27 to 60 from one frame to the next, highlighting temporal variations.	
95–162	Deviation of Features 27–94	Calculates the standard deviation for features 27 to 94 across all analysis windows within a signal, reflecting the fluctuation and stability of these features throughout the signal.	
163	Zero Crossing Rate	The frequency at which a signal changes sign.	numpy
164	Root Mean Square Energy	The square root of the average power of a signal.	
165	Slope Sign Changes Ratio	The rate at which the slope of a signal changes sign.	
168	Mean	The average value of the signal.	
169	Variance	The measure of the signal’s spread from its mean.	
170	Standard Deviation	The square root of the variance, which indicates the dispersion of a dataset.	
171	Interquartile Range	Difference between the 75th and 25th percentiles, indicating variability.	
166	Hjorth Mobility	Measure of the signal’s mean frequency or rate of change.	PyEEG
167	Hjorth Complexity	The ratio indicating the signal’s complexity compared to a sine wave.	
172	Detrended Fluctuation Analysis	The long-term correlation properties of a signal.	

**Table 8 biomimetics-09-00290-t008:** Synthesis of the different models and hyperparameters used for the classification task.

Model Name	Parameter	Values
AdaBoost	n_estimators	50, 100
	Learning Rate	0.1, 1
Extra Trees	n_estimators	100, 200
	Max Depth	None, 20
	Min Samples Split	2, 5
Gaussian NB	Var Smoothing	1 $\times \;10^{-9}$, 1 $\times \;10^{-8}$, 1 $\times \;10^{-10}$
Gradient Boosting	n_estimators	100, 200
	Learning Rate	0.1, 0.5
	Max Depth	3, 5
K-Neighbors	n_neighbors	5, 10, 15
	Weights	uniform, distance
LGBM	n_estimators	100, 200
	Learning Rate	0.1, 0.05
	Num Leaves	31, 64
Random Forest	n_estimators	100, 200, 300
	Max Depth	None, 10, 20
	Min Samples Split	2, 5
XGB	n_estimators	100, 200
	Learning Rate	0.1, 0.05
	Max Depth	3, 6

**Table 9 biomimetics-09-00290-t009:** Kruskal–Wallis test of the feature differences between control and eurythmy groups.

Feature	Control Avg	Eurythmy Avg	*p*-Value
slope_sign_changes_ratio	0.0297	0.0087	0.00
mfcc_9_mean	0.0534	0.0431	1.81 $\times \;10^{-239}$
mfcc_12_mean	0.0426	0.0344	5.81 $\times \;10^{-225}$
mfcc_13_mean	0.0303	0.0241	2.29 $\times \;10^{-207}$
mfcc_11_mean	0.0400	0.0322	6.01 $\times \;10^{-206}$
delta_spectral_spread_std	0.0632	0.0766	8.10 $\times \;10^{-135}$
dfa	1.6961	1.6382	1.89 $\times \;10^{-123}$
delta_energy_std	0.0509	0.0433	1.66 $\times \;10^{-35}$
variance	0.6927	0.4800	8.63 $\times \;10^{-33}$
interquartile_range	0.8215	0.6964	6.49 $\times \;10^{-27}$
energy_mean	0.1320	0.1526	2.93 $\times \;10^{-25}$
cepstra_8_std	0.7639	0.7646	2.38 $\times \;10^{-23}$
delta_chroma_8_std	4.51 $\times \;10^{-5}$	4.13 $\times \;10^{-5}$	6.85 $\times \;10^{-17}$
delta_mfcc_12_std	0.1189	0.1205	5.45 $\times \;10^{-9}$
mean	−0.0001	−0.0366	1.95 $\times \;10^{-8}$
delta_zcr_mean	−3.15 $\times \;10^{-7}$	7.33 $\times \;10^{-8}$	2.22 $\times \;10^{-6}$
delta_chroma_4_mean	2.49 $\times \;10^{-8}$	2.47 $\times \;10^{-8}$	3.38 $\times \;10^{-6}$
chroma_std_mean	0.0551	0.0550	8.27 $\times \;10^{-6}$
delta_energy_entropy_mean	2.60 $\times \;10^{-5}$	−3.41 $\times \;10^{-5}$	1.01 $\times \;10^{-5}$
delta_chroma_3_mean	1.04 $\times \;10^{-6}$	1.19 $\times \;10^{-6}$	3.20 $\times \;10^{-5}$
delta_spectral_rolloff_mean	−2.56 $\times \;10^{-8}$	1.32 $\times \;10^{-7}$	0.0003
cepstra_1_std	1.4264	1.4421	0.0017
delta_spectral_centroid_std	0.0379	−0.0376	0.0023
delta_chroma_std_mean	1.01 $\times \;10^{-6}$	−6.95 $\times \;10^{-7}$	0.0033
energy_std	0.0833	0.0819	0.0083

**Table 10 biomimetics-09-00290-t010:** Performance metrics for models in classifying eurythmy and control samples.

Model	F1	Accuracy	Precision	Recall
baseline	-	0.5038	-	-
adaboost	0.7134	0.7153	0.7211	0.7153
extratrees	0.7251	0.7257	0.7277	0.7257
gaussiannb	0.4531	0.5260	0.5558	0.5260
gradientboosting	0.7389	0.7396	0.7422	0.7396
kneighbors	0.5179	0.5179	0.5179	0.5179
**lgbm**	**0.7488**	**0.7494**	**0.7521**	**0.7494**
randomforest	0.7337	0.7344	0.7367	0.7344
xgb	0.7381	0.7390	0.7422	0.7390

**Table 11 biomimetics-09-00290-t011:** Statistical significances of Kruskal–Wallis between letter groups.

Feature	A Avg	G Avg	D Avg	*p*-Value
mean	0.0757	−0.0056	−0.0448	2.59 $\times \;10^{-30}$
root_mean_square_energy	0.7505	0.7186	0.6903	1.40 $\times \;10^{-9}$
cepstra_4_avg	0.2641	0.2623	0.2582	2.23 $\times \;10^{-9}$
cepstra_1_std	1.3481	1.3422	1.3313	9.00 $\times \;10^{-9}$
zero_crossing_rate	4.70 $\times \;10^{-5}$	4.71 $\times \;10^{-5}$	4.89 $\times \;10^{-5}$	0.0011
dfa	1.5778	1.5745	1.5777	0.0040
delta_chroma_3_std	0.0140	0.0142	0.0137	0.0069
spectral_spread_std	0.0356	0.0362	0.0357	0.0073
slope_sign_changes_ratio	0.0096	0.0102	0.0096	0.0082
spectral_flux_std	0.0719	0.0727	0.0710	0.0113
spectral_centroid_std	0.0235	0.0238	0.0235	0.0116
mfcc_3_mean	0.2130	0.2112	0.2132	0.0118
delta_mfcc_5_std	0.0944	0.0953	0.0952	0.0123
delta_spectral_spread_std	0.0621	0.0632	0.0624	0.0129
cepstra_2_std	1.2621	1.2697	1.2412	0.0198
mfcc_5_mean	0.1373	0.1362	0.1374	0.0231
mfcc_8_mean	0.0784	0.0782	0.0791	0.0317
mfcc_7_mean	0.0769	0.0765	0.0776	0.0340
delta_spectral_centroid_std	0.0389	0.0394	0.0390	0.0387
mfcc_6_mean	0.1447	0.1438	0.1450	0.0500

**Table 12 biomimetics-09-00290-t012:** Performance metrics for models in classifying eurythmy gestures.

Model	F1	Accuracy	Precision	Recall
baseline	-	0.3467	-	-
adaboost	0.4388	0.4392	0.4415	0.4392
extratrees	0.4349	0.4392	0.4363	0.4392
gaussiannb	0.4186	0.4185	0.4192	0.4185
**gradientboosting**	**0.4536**	**0.4586**	**0.4539**	**0.4586**
kneighbors	0.3738	0.3757	0.3750	0.3757
lgbm	0.4320	0.4350	0.4307	0.4350
randomforest	0.4444	0.4475	0.4435	0.4475
xgb	0.4534	0.4558	0.4530	0.4558

**Table 13 biomimetics-09-00290-t013:** Statistical analysis of feature differences across plant species.

Feature	Lettuce Avg	Tomato Avg	Basil Avg	*p*-Value
dfa	1.6519	1.6271	1.6783	0.00
slope_sign_changes_ratio	0.0220	0.0152	0.0260	3.54 $\times \;10^{-198}$
delta_mfcc_3_std	0.1085	0.1136	0.1062	1.15 $\times \;10^{-118}$
mfcc_6_std	0.0531	0.0554	0.0526	1.11 $\times \;10^{-105}$
mfcc_5_std	0.0510	0.0531	0.0505	1.92 $\times \;10^{-95}$
zcr_mean	0.0004	0.0004	0.0007	4.92 $\times \;10^{-91}$
mfcc_10_std	0.0472	0.0494	0.0470	3.37 $\times \;10^{-83}$
energy_entropy_std	0.1260	0.1342	0.1303	9.85 $\times \;10^{-83}$
delta_mfcc_8_std	0.0837	0.0870	0.0827	7.63 $\times \;10^{-80}$
delta_mfcc_7_std	0.0795	0.0828	0.0790	5.93 $\times \;10^{-75}$
variance	0.1940	0.1602	0.1780	1.48 $\times \;10^{-74}$
delta_mfcc_12_std	0.0729	0.0762	0.0726	2.25 $\times \;10^{-74}$
mfcc_9_std	0.0463	0.0484	0.0462	2.65 $\times \;10^{-74}$
mfcc_13_std	0.0385	0.0404	0.0389	4.10 $\times \;10^{-72}$
delta_mfcc_11_std	0.0770	0.0804	0.0764	9.50 $\times \;10^{-72}$
delta_energy_std	0.0788	0.0833	0.0820	2.63 $\times \;10^{-65}$
delta_chroma_11_std	0.0025	0.0027	0.0025	1.76 $\times \;10^{-61}$
delta_spectral_rolloff_std	0.0040	0.0043	0.0042	1.23 $\times \;10^{-57}$
delta_chroma_2_std	7.66 $\times \;10^{-5}$	8.14 $\times \;10^{-5}$	7.80 $\times \;10^{-5}$	7.79 $\times \;10^{-56}$
delta_chroma_3_std	0.0131	0.0139	0.0137	5.93 $\times \;10^{-55}$
delta_spectral_flux_std	0.1126	0.1178	0.1108	3.45 $\times \;10^{-51}$
cepstra_1_std	1.3283	1.3552	1.3269	2.29 $\times \;10^{-50}$
interquartile_range	0.3874	0.3811	0.4151	1.00 $\times \;10^{-49}$
energy_mean	0.2299	0.2210	0.2145	1.10 $\times \;10^{-43}$
cepstra_4_avg	0.2565	0.2648	0.2555	7.12 $\times \;10^{-38}$
spectral_spread_std	0.3321	0.3032	0.3156	1.14 $\times \;10^{-37}$
root_mean_square_energy	0.7415	0.7562	0.7344	4.99 $\times \;10^{-37}$
energy_std	0.0612	0.0632	0.0623	2.83 $\times \;10^{-32}$
hjorth_mobility	0.0001	0.0001	0.0001	4.94 $\times \;10^{-32}$
delta_spectral_flux_mean	0.0048	0.0051	0.0050	3.33 $\times \;10^{-26}$
hjorth_complexity	11,097.4615	11,059.9986	11,611.5156	5.48 $\times \;10^{-17}$
chroma_std_mean	0.0550	0.0550	0.0552	2.80 $\times \;10^{-14}$
mean	0.0076	0.0029	0.0126	7.16 $\times \;10^{-6}$
mfcc_10_mean	0.0568	0.0574	0.0565	6.16 $\times \;10^{-5}$
mfcc_11_mean	0.0382	0.0388	0.0385	0.0022
mfcc_3_mean	0.2311	0.2326	0.2317	0.0169
mfcc_9_mean	0.0514	0.0519	0.0514	0.0204
delta_mfcc_5_mean	3.17 $\times \;10^{-5}$	−7.96 $\times \;10^{-5}$	1.78 $\times \;10^{-5}$	0.0241
delta_mfcc_8_mean	4.80 $\times \;10^{-5}$	−3.96 $\times \;10^{-5}$	−1.37 $\times \;10^{-5}$	0.0394

**Table 14 biomimetics-09-00290-t014:** Model evaluation metrics.

Model	F1	Accuracy	Precision	Recall
baseline	-	0.3605	-	-
adaboost	0.4136	0.4180	0.4186	0.4180
extratrees	0.4225	0.4299	0.4355	0.4299
gaussiannb	0.2759	0.3670	0.3870	0.3670
gradientboosting	0.4949	0.4981	0.5005	0.4981
kneighbors	0.3382	0.3515	0.3443	0.3515
lgbm	0.4849	0.4868	0.4881	0.4868
randomforest	0.4474	0.4534	0.4601	0.4534
**xgb**	**0.5051**	**0.5084**	**0.5108**	**0.5084**

## Data Availability

The results, code, charts, and models from this study are available openly at https://github.com/alvaro-francisco-gil/Plant-Reactivity-Analysis (accessed on 6 April 2024).

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
