# Peer review of "Can Plants Perceive Human Gestures? Using AI to Track Eurythmic Human–Plant Interaction"

_biomimetics, 2024, doi:10.3390/biomimetics9050290_

Round 1
Reviewer 1 Report
Comments and Suggestions for Authors
Gloor et al. propose a method aimed at investigating whether plants can respond to human movement by altering their electrical signals. The authors used plant voltage measurement devices and machine learning to observe whether changes in plant voltage over time can predict harmonious movements and what type of harmonious gestures were performed. I recommend acceptance of the manuscript after minor revision as exemplified below:
(1) The use of a limited range of three plant species fails to address broader issues in the manuscript.
(2) Please explain why the measurements were inconsistent for lettuce, basil, and tomato in Section 4.1.
(3) Data points in Figure 5 are unclear, making it difficult to distinguish the specific electrical signal characteristics of a certain species.
(4) Please explain the use cases and features of these three libraries: Scikit-learn, LightGBM, and XGBoost.
(5) The format of the References is inconsistent.
Reviewer 2 Report
Comments and Suggestions for Authors
The topic of this paper is pretty interesting, as it utilizes AI to track human-plant interaction. The overall structure of the paper is good, and current results can support the author's three hypotheses to a certain degree. The following are my main comments about the manuscript:
1. The motivation and contribution of the paper are not very clear to me. The authors should clarify them in the introduction section.
2. Regarding the related work section, the authors don't point out the research gap in the current work on the plant sense domain. Please explain more about the current research gaps.
3. Section 4.1 shows four generations of lettuce and two generations of both basil and tomato. Why lettuce has two more generations? I assume the variable should remain the same for comparison purposes.
4. In section 4.3, the authors mention they conducted several experiments to decide the window size and hop length values. Please add more details about the experiment.
5. In the results section, the authors compare the performance among eight different ML models and baseline method; what is baseline method?
Comments on the Quality of English LanguageThe overall quality of English writing is good and only grammar of some sentences need to be improved.
